# Early Hominin Dispersal across the Qinling Mountains, China, during the Mid-Pleistocene Transition

**Xiaoqi Guo [1], Xuefeng Sun [1,*], Huayu Lu [1], Shejiang Wang [2,*] and Chengqiu Lu [3]**

1. School of Geography and Ocean Science, Nanjing University, Nanjing 210023, China; 602022270012@smail.nju.edu.cn (X.G.); huayulu@nju.edu.cn (H.L.)
2. Key Laboratory of Vertebrate Evolution and Human Origins, Institute of Vertebrate Paleontology and Paleoanthropology, Chinese Academy of Sciences, Beijing 100044, China
3. Hubei Provincial Institute of Cultural Relics and Archaeology, Wuhan 430077, China; luchengqiu@hotmail.com
* Correspondence: xuefeng@nju.edu.cn (X.S.); wangshejiang@ivpp.ac.cn (S.W.)

**Abstract:** The Qinling Mountain Range (QMR), where more than 500 hominin fossils and Paleolithic sites have been preserved, was a major center of hominin evolution and settlement and an important link for the hominin migration and dispersal between the north and the south during the Pleistocene in China. The rich culture remains and the related data make it possible and meaningful to study the characteristics and mechanisms of hominin occupation and dispersal in the region. This paper has summarized and analyzed the geographical distributions and chronologies of 55 dated hominin fossils and Paleolithic sites in the QMR to date. By combining them with the evidence from the loess–paleosol sequence, a relatively continuous and chronological sequence of hominin occupation and dispersal has been established, in which we have identified five stages, viz. ~before 1.2 Ma, the sporadic occurrence stage of early hominin occupation; ~1.2–0.7 Ma, the initial expansion stage; ~0.7–0.3 Ma, the stability and maintenance stage; ~0.3–0.05 Ma, the large-scale expansion stage; ~0.05–0.01 Ma, the sharp decline stage of the record of hominin occupation. We conclude that the environmental and ecosystem changes associated with the MPT drove early hominins to disperse southwards across the QMR. In addition, the evidence suggests that the hominin occupation and dispersal here was broadly continuous during both glacial and interglacial scales from early to late Pleistocene, and that the southern QMR provided a glacial refuge.

**Keywords:** Qinling Mountain Range; mid-Pleistocene transition; hominin occupation and dispersal; chronological framework; dispersal corridors





## 1. Introduction

The role of climate change in pacing hominin evolution and dispersals has been the subject of intense study and debate [1–7]. Some studies have suggested that climate shifts were a key factor in driving hominin species' distributions and dispersals [7–12]. In China, intensified hominin activity gradually shifted southward under the influence of multiple glacial–interglacial cycles [13]. Norton et al. (2011) [14] have preliminarily proposed that there were at least two possible migration corridors between North and South China during the Pleistocene. Based on that, and in combination with the distribution of the Paleolithic sites from Early to Middle Pleistocene in China, we have further refined these two corridors, as shown in Figure 1. One of the migration corridors primarily ran along the eastern side of the China's third ladder, which is composed of plain and hilly landforms with an average elevation of less than 500 m. The hominins could migrate along the more open and low-lying areas on the corridor. Another important migration corridor mainly followed the eastern side of the second ladder, which includes the landforms such as plateaus, mountains, and basins with an average elevation of 1000–2000 m. The hominins were able to move through the Qinling Mountain Range (QMR). And substantial

archaeological evidence indicates that these depressions in the basins of the QMR were likely interconnected throughout the Pleistocene, and thus represent an important corridor for hominin dispersal and migration throughout central China [7,15–19]. Obviously, the QMR is in a critical position for hominin migration and dispersal.

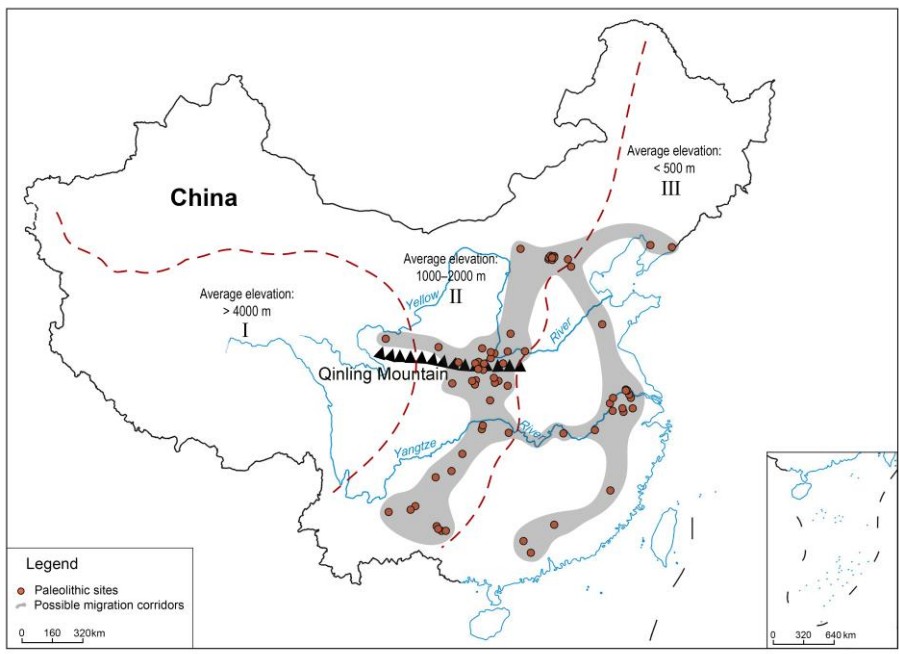

**Figure 1. Possible migration and dispersal corridors of early hominins based on the distribution of the archaeological sites from Early to Middle Pleistocene in China**. (The Paleolithic sites of Early and Middle Pleistocene are cited from Lu et al. (2021) [20]).

In the QMR, over 500 Paleolithic sites, several hominin fossils, and more than 200,000 lithic artifacts have been discovered, making it one of the centers of early hominin settlement and evolution in the Chinese Paleolithic [15,16,21–31]. Since the discovery of hominin skull fossils at the Gongwangling site in the 1960s [21], there has been a strong focus on the Paleolithic archaeology of the QMR. And new hominin fossils and Paleolithic sites have been discovered and studied one after another, such as Longgangsi-3 (1.2–0.71 Ma [32]), the Yunxian site (0.936 Ma [33], or 0.8–0.785 Ma [34], or 1.0–0.78 Ma [35]), Longyadong Cave (0.39–0.27 Ma [36]), and so on. Paleolithic assemblages in the QMR include the typical southern cobble/pebble tools industry and northern small-core- lake-retouched flake tool industry, and these diversified assemblages demonstrate that the QMR was an important region in terms of hominin evolution and cultural exchange between southern and northern China [15,16,30]. Furthermore, with the continuous development of dating techniques on hominin fossils and Paleolithic sites, more and more age data provide effective support for establishing a chronological framework of hominin occupation, migration, and dispersal [7,32–52].

All in all, the QMR was not an obstacle to hominin migration and diffusion between the north and south [17], but a "refuge" for the hominins during the glacial period [7] and an important link for the hominin migration and dispersal between the north and south. Therefore, it is necessary to clarify the spatiotemporal features and driving factors of hominin occupation, migration, and dispersal in the area. This article focuses on the chronological sequence and spatial distribution of human occupation in the QMR, aiming to build the time framework of hominin occupation, migration, and diffusion in the region, as well as exploring the driving factors and mechanisms involved.

## 2. Materials and Methods

### 2.1. Geographical Setting

The Qinling Mountain Range, located between 32° N and 35° N, forms a natural barrier separating the Chinese Loess Plateau in North China and the tropical and subtropical forests in South China [14,52–54]. The climate varies from arid to semi-arid and semi-humid. The northern QMR is characterized by a temperate monsoon and temperate continental climate, while the southern and eastern QMR are characterized by a subtropical monsoon climate [55].

The landform of the QMR is mainly characterized by a mountain-basin system. Several fault basins have formed since the Cenozoic period, including the Hanzhong, Ankang, and Luonan basins, etc. [56,57], and more than five river terraces were formed over the past ~1.2 Ma BP [17]. In addition, a large area of loess sediments mantled the fluvial terraces and mountain tops, with thicknesses of several tens to more than two hundred meters [17]. Loess deposits in the northern QMR are thick, and typical loess–paleosol sequences are developed, which are relatively continuous loess deposits on the scale of the glacial–interglacial period [15]. The loess deposits in the southern and the eastern QMR are slightly different from the typical loess in the Loess Plateau, which are between the typical loess and Xiashu loess; the loess is also relatively thinner, the grain size is finer, and the color is considerably more red [18,26–29,44,46–48].

These loess–paleosol sequences are important carriers for recording the changes in the Paleoenvironment [58], and they have retained rich hominin fossils and Paleolithic cultural remains. The study of loess is closely linked with Paleolithic archaeology, which can provide environmental background and chronological sequence for investigating hominin occupations and evolutions [59,60]. The research progress of the 55 hominin fossils and Paleolithic sites that have independent dating data in the QMR and surrounding areas are summarized in this paper (The distribution of these 55 sites is shown in Figure 2), and among them, 44 sites were systematically dated by our team, and 15 sites were dated by other teams. Most of these 55 hominin fossils and Paleolithic sites were found in loess deposits developed in the northern, southern, and eastern QMR, and these lithic culture remains are densely distributed in the watersheds of the Bahe River, Hanjiang River and South Luohe River.

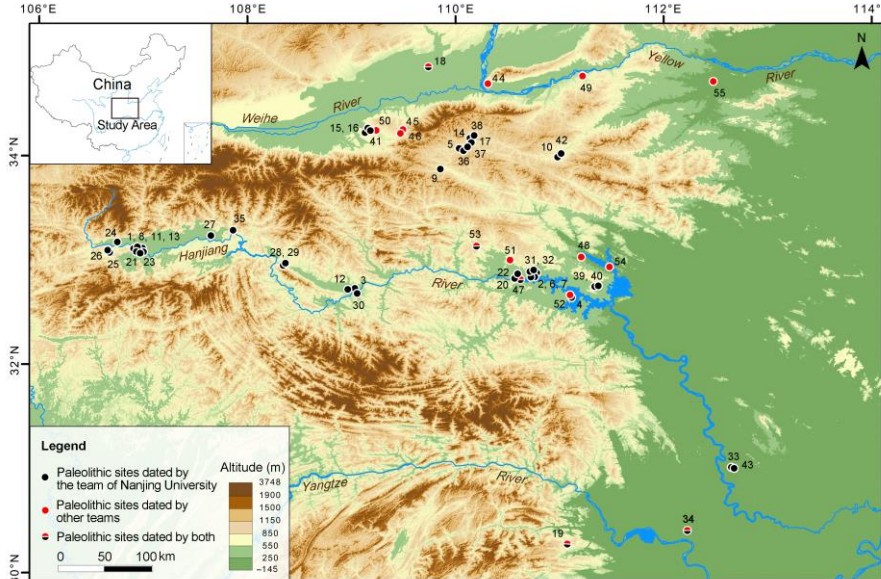

**Figure 2. Distribution of dated hominin fossils and Paleolithic sites in the QMR and surrounding areas**. The black dots represent the sites dated by our team, the red dots represent the research of other teams, the half-red and half-black dots represent the sites dated by both: 1. Longgangsi-3;

2. Yuelianghu; 3. Wutaicun; 4. Guanmenyan; 5. Shangbaichuan; 6. Wujiagou; 7. Liubeijiuchang; 8. Longgangsi-2; 9. Miaokou; 10. Qiaojiayao; 11. Yaochangwan; 12. Luojiacun; 13. Longgangsi-1; 14. Liuwan; 15. Jijiawan; 16. Ganyu; 17. Longyadong Cave; 18. Dali; 19. Changyang Cave; 20. Houfang; 21. Hejialiang; 22. Dishuiyan; 23. Liangshancun; 24. Lianfeng-1; 25. Lianfeng-2; 26. Yangjiaping; 27. Erligou; 28. Chihe-1; 29. Chihe-2; 30. Qujiahe; 31. Jinkuangcun-2; 32. Jinkuangcun-3; 33. Luohansi-2; 34. Jigongshan; 35. Jinshuihekou; 36. Yanling; 37. Shizilukou; 38. Huaishuping; 39. Huoshiwa; 40. Houshanpo; 41. Diaozhai; 42. Zhuangzixun; 43. Luohansi-1; 44. Xihoudu; 45. Shangchen; 46. Gongwangling; 47. Yunxian; 48. Meipu; 49. Shuigou-huixinggou; 50. Chenjiawo; 51. Bailongdong Cave; 52. Shuangshu; 53. Huanglong Cave; 54. Maling 2A; 55. Beiyao.

### 2.2. Chronological Study of the 55 Hominin Fossils and Paleolithic Sites

The main methods of dating these 55 Paleolithic sites are optically stimulated luminescence (OSL, including single-aliquot regenerative-dose-based OSL (SAR-OSL), thermally transferred OSL (TT-OSL) and post-infrared infrared-stimulated luminescence (pIRIR)), paleomagnetism, and pedostratigraphic correlation with the typical loess–paleosol sequence, and some other independent dating methods are occasionally used, such as $^{26}Al/^{10}Be$ burial dating, U-series, ESR/U-series, and ESR, etc. Pedostratigraphic correlation, paleomagnetism, and OSL are the three most commonly used methods. Paleomagnetism is applicable to eolian loess, fluvial–lacustrine sediments, and cave sediments, among various others. It can only obtain accurate results of geomagnetic polarity inversion boundaries (e.g., the Brunhes/Matuyama (B/M) boundary, Jaramillo, Olduvai, etc.), and its dating accuracy of layers between polarity inversion boundaries is limited [39,61]. The materials of OSL dating method are mainly quartz and feldspar grains purified from loose deposits. The upper dating limit is generally 0.06–0.1 Ma for SAR-OSL [62–64], 0.78 Ma for TT-OSL [48,65], and 0.5 Ma for pIRIR [66]. However, the dating ability of the TT-OSL and pIRIR methods in practical application generally does not exceed 0.2–0.3 Ma [32]. In general, the hominin fossils and Paleolithic sites during the Early and Middle Pleistocene are mainly dated via paleomagnetism and pedostratigraphic correlation, and the sites during the Late-Middle Pleistocene and Late Pleistocene are mainly dated via OSL, and U-series and other dating methods are mostly used for dating cave sites. All chronological results of the 55 hominin fossils and Paleolithic sites are summarized in Table 1.

**Table 1.** The chronological results of the 55 hominin fossils and Paleolithic sites in the QMR.

| Location | Site | Age (Ma) | Dating Method | Loess–Paleosol Period | MIS | References |
|---|---|---|---|---|---|---|
| | Shangchen | 2.12–1.26 | PM, PC | S15–S27 | 39–81 | [67] |
| | | 1.15 | PM, PC | | | [37] |
| | Gongwangling | 1.63 | PM, PC | L15/S22–S23 | 12/53–56 | [68] |
| | | 1.82 | $^{26}Al/^{10}Be$ | | | [49] |
| | | 1.27 | PM, PC | | | [40] |
| | Xihoudu | 1.4 | $^{26}Al/^{10}Be$ | S17–S18 | 43–45 | [69] |
| | | 2.43 | $^{26}Al/^{10}Be$ | | | [52] |
| | Chenjiawo | 0.65 | PM, PC | S6 | 17 | [37,70] |
| Northern QMR | Shuigou-Huixinggou | 0.9 | PM, PC | L9 | 23 | [71] |
| | Jijiawan | 0.43–0.38 & 0.33–0.30 & 0.25–0.19 & 0.13–0.07 | OSL, pIRIR, PM, PC | S1, S2, S3, S4 | 5, 7, 9, 11 | [72] |
| | Ganyu | 0.43–0.38 & 0.13–0.07 | pIRIR, PC | S1, S4 | 5, 11 | [72] |
| | | 0.23–0.18 | U | | | [73] |
| | | 0.27 | PC | | | [43] |
| | Dali | 0.30–0.26 | IRSL, ESR, U | L3 | 8 | [74] |
| | | 0.28 | ESR/U | | | [75] |
| | | 0.26–0.27 | TT-OSL, pIRIR | | | [48] |
| | Diaozhai | 0.07–0.02 | OSL, pIRIR | L1 | 2–4 | [72] |

**Table 1.** *Cont.*

| Location | Site | Age (Ma) | Dating Method | Loess–Paleosol Period | MIS | References |
|---|---|---|---|---|---|---|
| Eastern QMR | Shangbaichuan | 0.8–0.7 & 0.4–0.3 & 0.2–0.1 | OSL, PM, PC | S8 | 21 | [18,44] |
| | Miaokou | 0.7–0.6 | OSL, PM, PC | S5–S6 | 15–17 | [19] |
| | Qiaojiayao | 0.62–0.6 | OSL, PM, PC | S5 | 15 | [45] |
| | | 0.4–0.3, 0.2–0.1 | OSL, PM, PC | | | [18,44] |
| | Liuwan | 0.625–0.581 & 0.575–0.568 | PM, PC | S5 | 15 | [47] |
| | | >0.60 | $^{26}$Al/$^{10}$Be | | | [76] |
| | Longyadong Cave | 0.5–0.25 | TL | L3–S4 | 9–11 | [24] |
| | | 0.39–0.27 | TT-OSL | | | [36] |
| | Yanling | 0.23–0.07 | pIRIR | S1 | 5 | [7,77] |
| | Beiyao | 0.2–0.01 | PC, OSL, TT-OSL,$^{14}$C | S0–S2 | 1–7 | [78] |
| | Shizilukou | 0.12–0.04 | TT-OSL, pIRIR | L1–S1 | 3–5 | [7,79] |
| | Huaishuping | 0.09–0.01 | OSL | L1–S1 | 2–5 | [7,80] |
| | Zhuangzicun | 0.07 | OSL | L1 | 4 | [26] |
| Southern QMR | Yunxian | 0.87–0.83 | PM, PC | S8–S9 | 21–26 | [81] |
| | | 0.936 | PM, PC | | | [33] |
| | | 0.581 | ESR | | | [82] |
| | | 0.8–0.785 | PM, PC | | | [34] |
| | | 1.0–0.78 | PM, PC | | | [35] |
| | | 1.1 | ESR/U | | | [83] |
| | Meipu | 0.99–0.78 | U, PM, PC | L8–S9 | 20–26 | [84] |
| | | 0.76 | $^{26}$Al/$^{10}$Be | | | [85] |
| | Bailong Cave | 0.78 | PM, PC | S5/S7 | 13/19 | [86] |
| | | 0.5 | ESR/U | | | [50] |
| | Huanglong Cave | 0.095 & 0.079 | U | S1 | 5 | [87] |
| | | 0.103 & 0.044 | U | | | [88] |
| | | 0.1–0.057 | U | | | [89] |
| | | 0.101–0.081 | U | | | [51] |
| | | <0.001 | $^{14}$C | | | [90] |
| | Changyang Cave | 0.195 | U | L2 | 6 | [91–93] |
| | | 0.196–0.143 | U | | | [94] |
| | Longgangsi-3 | 1.2–0.71 | pIRIR, PM, PC | L7–S14 | 18–37 | [32] |
| | Yuelianghu | 0.99–0.94 & 0.87–0.81 | PM, PC | S8 | 21 | [95] |
| | Wutaicun | 0.98–0.95 | pIRIR, PM, PC | S9 | 26 | [32] |
| | Guanmenyan | 0.8 | PM, PC | L8 | 20 | [95] |
| | Liubeijiuchang | 0.78–0.71 & 0.15–0.13 | pIRIR, PM, PC | S1, S7 | 5,19 | [96] |
| | Wujiagou | 0.78–0.71 | PM, PC | S7 | 19 | [96] |
| | Longgangsi-2 | 0.78 & 0.7 | pIRIR, PM, PC | S6 | 17 | [32] |
| | Shuangshu | 0.65–0.52 | ESR | S5 | 13–15 | [97] |
| | Yaochangwan | 0.6 | OSL, PM, PC | S5 | 15 | [32,46] |
| | Luojiacun | 0.6 | pIRIR, PM, PC | S5 | 15 | [32,46] |
| | Longgangsi-1 | 0.58–0.07 | PC | S1–S5 | 5–15 | [7] |
| | Maling 2A | 0.39–0.22 | pIRIR | S2–L4 | 7–10 | [98] |
| | Jinshuihekou | >0.15 | pIRIR | L2- | 6- | [99] |
| | Liangshancun | 0.19–0.14 | TT-OSL | L2 | 6 | [100] |
| | Lianfeng-2 | 0.17–0.11 | TT-OSL | S1–L2 | 5–6 | [100] |
| | Jinkuangcun-2 | 0.17–0.08 | TT-OSL | S1–L2 | 5–6 | [100] |
| | Chihe-2 | 0.16–0.11 | TT-OSL | S1–L2 | 5–6 | [100] |
| | Houfang | 0.15–0.8 | TT-OSL | S1 | 5 | [30,35] |
| | Chihe-1 | 0.15–0.11 | TT-OSL | S1–L2 | 5–6 | [100] |
| | Jinkuangcun-3 | 0.15–0.08 | TT-OSL | S1–L2 | 5–6 | [100] |
| | Lianfeng-1 | 0.15–0.07 | TT-OSL | L1, L2 | 4, 6 | [100] |
| | Erligou | 0.13–0.11 | TT-OSL | S1 | 5 | [100] |
| | Qujiahe | 0.13–0.1 | TT-OSL | S1–L2 | 5–6 | [100] |
| | Yangjiaping | 0.12–0.07 | TT-OSL | S1 | 5 | [100] |
| | Dishuiyan | 0.11–0.05 | OSL, TT-OSL | L1–S1 | 4–5 | [35] |
| | Hejialiang | 0.09–0.07 | TT-OSL | S1 | 5 | [46] |
| | Houshanpo | >0.07 | OSL | S1– | 5- | [101] |

**Table 1.** *Cont.*

| Location | Site | Age (Ma) | Dating Method | Loess–Paleosol Period | MIS | References |
|---|---|---|---|---|---|---|
| | Huoshiwa | >0.05 | OSL | L1– | 3- | [101] |
| | Luohansi-2 | 0.18–0.11 | TT-OSL | S1–L2 | 5–6 | [102] |
| Southern QMR | Luohansi-1 | 0.05–0.03 | OSL | L1 | 3–4 | [102] |
| | Jigongshan | 0.02–0.01 & >0.05 | $^{14}$C | | | [103,104] |
| | | 0.15–0.11 & 0.04–0.02 | OSL, TT-OSL | L1, S1–L2 | 3, 5–6 | [105] |

Note: PC, pedostratigraphic correlation with the loess-paleosol sequence; PM, paleomagnetism; OSL, optically stimulated luminescence; TT-OSL, thermally transferred OSL; pIRIR, post-infrared infrared stimulated luminescence; $^{26}$Al/$^{10}$Be, $^{26}$Al/$^{10}$Be burial dating; U, Uranium-series dating or U-series; ESR, electron spin resonance; TL, thermoluminescence; IRSL, infrared stimulated luminescence; $^{14}$C, AMS $^{14}$C dating; MIS, marine isotope stage [106]; The corresponding loess-paleosol periods were referenced by Ding et al. [107].

### 2.2.1. Northern QMR

There are relatively few sites that have been dated independently in the northern QMR, including three hominin fossil sites and six Paleolithic sites.

The sites belonging to the Early Pleistocene include the Gongwangling, Shangchen, and Xihoudu. The age of the Gongwangling human fossil site has been studied many times: the paleomagnetic age measured in the 1980s is about 1.15 Ma, the fossil layer corresponds to the middle of the L15 layer [37], the latest paleomagnetic age is about 1.63 Ma, the fossil layer is updated to the S22–S23 layer [68], and the latest $^{26}$Al/$^{10}$Be burial age is less than 1.82 Ma [49]. The paleomagnetic age of the Shangchen site is about 2.12–1.26 Ma [67]. The dating of the Paleolithic remains at the Xihoudu site has been ongoing, although controversial. The early paleomagnetic age is about 1.27 Ma [40]. The later $^{26}$Al/$^{10}$Be burial age is about 1.4 Ma [69] The latest $^{26}$Al/$^{10}$Be burial dating pushed its age back to about 2.43 Ma [52], again redefining the earliest record of human activities in China. However, it is too different from the previous age, so the reliability still needs further verification.

Regarding the Middle Pleistocene, the Chenjiawo human fossil site is dated to about 0.65 Ma by paleomagnetism [37,70]; the age of the Dali human fossil site is dated to about 0.23–0.18 Ma via U-series dating [73], to about 0.27 Ma via pedostratigraphic correlation [43], to about 0.26–0.30 Ma by the comprehensive dating of IRSL, ESR, and U-series dating [74], to about 0.28 Ma via ESR/U-series [75], and to about 0.26–0.27 Ma via TT-OSL and pIRIR dating from our team [48]. The Shuigou-Huixinggou site in Sanmenxia basin is dated to about 0.9 Ma via paleomagnetism [71]; the four cultural layers of the Jijiawan site are dated to about 0.43–0.38, 0.33–0.30, 0.25–0.19, and 0.13–0.07 Ma, respectively, and the two cultural layers of the Ganyu site are dated to about 0.43–0.38 and 0.13–0.07 Ma, respectively, via the comprehensive dating of OSL, pIRIR, paleomagnetism, and pedostratigraphic correlation [72].

Regarding the Late Pleistocene, the age of the Diaozhai site is about 0.07–0.02 Ma according to the comprehensive dating of OSL, pIRIR, paleomagnetism, and pedostratigraphic correlation [72].

### 2.2.2. Eastern QMR

The sites that have been dated in the eastern QMR include one hominin fossil site and nine Paleolithic sites. Currently, no Paleolithic site from the Early Pleistocene has been discovered or dated.

The sites that are attributable to the Middle Pleistocene are one human fossil site (Longyadong Cave) and six Paleolithic sites. The earliest thermoluminescence age of the Longyadong Cave site is about 0.5–0.25 Ma [24], and the latest TT-OSL age is about 0.39–0.27 Ma [36]. However, there are still lower cultural layers (fluvial deposit, the earliest gravel layer) that have not been sampled, so it is impossible to obtain an independent age. The comprehensive ages derived from OSL, paleomagnetism, and pedostratigraphic correlation of the Shangbaichuan Paleolithic site are about 0.8–0.7 Ma, 0.4–0.3, and 0.2–0.1 Ma [18,44], while those of the Miaokou and Qiaojiayao Paleolithic sites are about

0.7–0.6 Ma [19] and 0.62–0.6 Ma [45], respectively. In the Liuwan site, the comprehensive dating results of OSL, paleomagnetism, and pedostratigraphic correlation are about 0.4–0.3 and 0.2–0.1 Ma [18,44]; the paleomagnetic and pedostratigraphic correlation ages of artifact layer 1 at localities 2 and 3, and layer 2 at locality 3 are about 0.625–0.581 and 0.575–0.568 Ma, respectively [47]; the $^{26}$Al/$^{10}$Be burial age of artifact layer 1 in locality 3 is over $0.60 \pm 0.12$ Ma [76]. The TT-OSL and pIRIR ages of the Yanling site are about 0.23–0.07 Ma [7,77]. The comprehensive dating results of pedostratigraphic correlation, OSL, TT-OSL, and $^{14}$C of the Beiyao site are about 0.2–0.01 Ma [78].

Regarding the Late Pleistocene, the TT-OSL and pIRIR ages of the Shizilukou site are about 0.12–0.04 Ma [7,79]; the OSL ages of Huaishuping and Zhuangzicun sites are about 0.09–0.01 Ma [7,80] and 0.07 [26] Ma, respectively.

### 2.2.3. Southern QMR

There are five hominin fossil sites have been dated in the southern QMR. Regarding the Early Pleistocene, the Yunxian site is dated to about 0.87–0.83 Ma [81] and 0.936 Ma [33] via paleomagnetism; to about 0.581 Ma [82] via ESR dating; to about 0.8–0.785 Ma [34] and 1.0–0.78 Ma [35] via pedostratigraphic correlation and paleomagnetism; and to about 1.1 Ma via the ESR/U-series joint dating [83]. And the Meipu site is dated to about 0.99–0.78 Ma via paleomagnetism [84]. Regarding the Middle Pleistocene, the early $^{26}$Al/$^{10}$Be burial age of the Bailong Cave site is 0.76 Ma [85]), the latest paleomagnetic age of the lower cultural layer is less than 0.78 Ma [86], and the ESR/U-series joint dating result date it to about 0.5 Ma [50]. In addition, the Huanglong Cave and Changyang Cave sites were occupied during the Late Pleistocene. In the Huanglong Cave site, the U-series ages of two rhinoceros teeth coeval with the hominin fossils are about 0.095 and 0.079 Ma [87], the U-series ages of a stalagmite is about 0.103 Ma, and the ESR age of a rhinoceros tooth is about 0.044 Ma [88]). The subsequent U-series age is about 0.1–0.057 Ma [89], the latest high-precision mass spectrometric U-series dating age is about 0.101–0.081 Ma [51], the latest $^{14}$C age is about less than 0.001 Ma [90]. The Changyang Cave site in western Hubei Province was the first early *Homo sapiens* fossil site found in southern China. In 1986, Yuan et al. (1986) [91] determined that the $^{230}$Th ages of two dicerorhinus teeth in the fossil layer were 0.196 and 0.194 Ma, respectively, and no scholar has studied the site since then. Therefore, the average age of 0.195 Ma is considered to be the age of the Changyang hominin fossil [92,93]. Furthermore, the latest U-Th age is about 0.196–0.143 Ma [94].

In total, 33 Paleolithic sites are dated in the southern QMR. During the Early Pleistocene, the comprehensive dating results of pIRIR, paleomagnetism, and pedostratigraphic correlation of the Longgangsi-3 and Wutaicun sites are about 1.2–0.71 and 0.98–0.95 Ma, respectively [32]; the paleomagnetic ages of the Yuelianghu, Guanmenyan, Wujiagou and Liubeijiuchang (the lower cultural layer) sites are about 1.0–0.8 Ma [95], 0.8 Ma [95], 0.8–0.7 Ma [96], and 0.8–0.7 Ma [96], respectively.

Regarding the Middle Pleistocene, the comprehensive dating results of pIRIR, paleomagnetism, and pedostratigraphic correlation of the Longgangsi-2 site date it to approximately 0.78 Ma for the lower artifact layer and approximately 0.70 Ma for the upper artifact layer [32]; the ESR age of the Shuangshu site is about 0.65–0.52 Ma [97]; the ages of the Yaochangwan and Luojiacun sites are both about 0.6 Ma [32,46]; the age of the Longgangsi-1 site is about 0.58–0.07 Ma [7]; and the pIRIR ages of the Maling 2A and Jinshuihekou sites are 0.39–0.22 Ma [98] and over 0.15 Ma [99], respectively. TT-OSL dating has also been used for eight sites, including the Houfang site (about 0.15–0.8 Ma) [30,35], Liangshancun site (about 0.19–0.14 Ma) [100], Lianfeng-1 site (about 0.15–0.07 Ma) [100], Lianfeng-2 site (about 0.17–0.11 Ma) [100], Chihe-1 site (about 0.15–0.11 Ma) [100], Chihe-2 site (about 0.16–0.11 Ma) [100], Jinkuangcun-2 site (about 0.17–0.08 Ma) [100], and Jinkuangcun-3 site (about 0.15–0.08 Ma) [100].

Regarding the Late Pleistocene, the age of the Qujiahe site is about 0.13–0.1 Ma [100], the Erligou site is about 0.13–0.11 Ma [100], the Yangjiaping site is about 0.12–0.07 Ma [100], the Dishuiyan site is about 0.11–0.05 Ma [35], and the Hejialiang site is about 0.09–0.07 Ma [46], all

of which have been dated via TT-OSL. And the OSL ages of the Huoshiwa and Houshanpo sites are >0.05 Ma and >0.07 Ma, respectively [101]. In addition to the above sites, the TT-OSL age of the Luohansi-2 site in Zhongxiang basin is about 0.18–0.11 Ma, and the OSL age of the Luohansi-1 site in Zhongxiang basin is about 0.05–0.03 Ma [102]. In the Jigongshan site, Peking University conducted preliminary $^{14}$C dating and found that the upper layer is approximately 0.002–0.001 Ma, whereas the lower layer is older than 0.005 Ma [103,104]; the latest TT-OSL age of the lower cultural layer is about 0.15–0.11 Ma, and the OSL age of the upper cultural layer is about 0.04–0.02 Ma [105].

## 3. Discussion

### 3.1. Chronological Framework of Hominin Occupation

According to the age statistics of the hominin fossils and the Paleolithic sites that have been dated in QMR and the pedostratigraphic correlation with the loess–paleosol sequence, we can see the patterns of hominin occupation in different periods in the Qinling Mountains and preliminarily establish a relatively continuous chronological sequence (Figure 3). The hominin occupation in the QMR was continuous in the loess–paleosol sequence the S27 to L1 layers during the Pleistocene. Before 1.2 Ma, the occurrence of early hominins was sporadic, and the time span of a single site was large. The Gongwangling site (~1.6 or 1.15 Ma) is the earliest hominin fossil site. Since 1.2–0.7 Ma, the number of the sites had gradually increased, and the intensity of the hominin occupation began to increase. Additionally, the earliest hominin occupations in the southern and eastern QMR occurred at the Longgangsi-3 site (~1.2 Ma) and the Shangbaichuan site (~0.8 Ma), respectively. During 0.7–0.3 Ma, the number of the Paleolithic sites kept increasing, and hominin occupations during the interglacial periods, S5 (MIS 13, 14, 15), S4 (MIS 11), and S3 (MIS 9), were relatively frequent. At about 0.30–0.10 Ma, the number of the sites had significantly increased on a large scale, and the intensity of the hominin occupation had increased dramatically, which was basically continuous in the S2 (MIS7), L2 (MIS 6), S1 (MIS 5), and L1 (MIS 2, 3, 4) layers. The strongest hominin occupations were found in the glacial period L2 (MIS 6) and the interglacial period S1 (MIS 5). After 0.05 Ma, there are few hominin fossils and Paleolithic sites that have been dated at present. Although some records of stone artifacts have also been found in the field survey, they are still relatively few in general. This may be linked to the decline in the gross primary productivity (GPP), which was driven by climate change [7], or related to the arrival of modern humans [90,108–111], or may be due to the hominin occupation areas, which, at that time, overlapped with those of the current occupants and were disturbed and destroyed by the current human activities. More in-depth research is needed.

### 3.2. Spatiotemporal Characteristics of Hominin Migration and Dispersal

Hominin occupation and diffusion in the QMR were basically continuous in glacial and interglacial scales during the Pleistocene, but there were differences in the northern, eastern, and southern QMR. From Figure 3, it can be preliminarily seen that the human occupation in the eastern and southern QMR were significantly later than that in the northern QMR, and it seems to be a certain correlation with the "Mid-Pleistocene transition" (MPT), which was an event in which the global climate underwent a significant change and is generally believed to have occurred between 1.2 and 0.7 Ma [112–114]. As a result, we cut across the Qinling Mountains along the Bahe River valley and Danjing River valley, and have marked relevant dated hominin fossils and Paleolithic sites as the time periods of >1.2 Ma (before MPT), 1.2–0.7 Ma (MPT), and 0.7–0.05 Ma (after MPT) (Figure 4), and we have tried to explore the spatiotemporal characteristics and the mechanism of the early hominin settlement, migration, and dispersal in QMR.

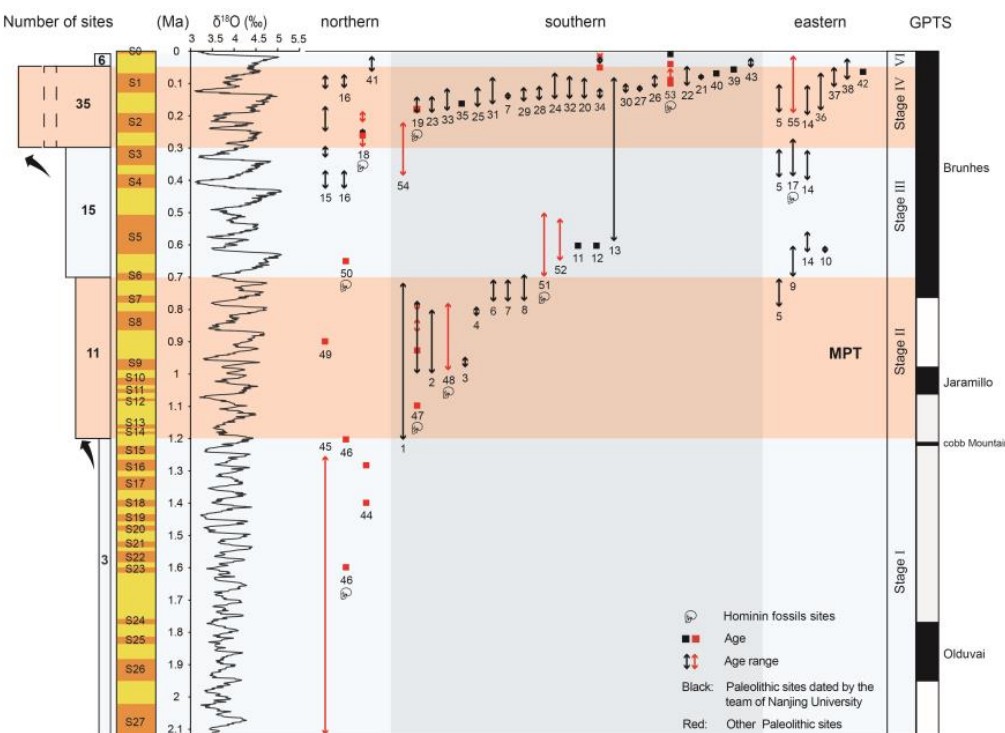

**Figure 3. Chronological framework of hominin occupation in the QMR**. (The black symbols indicate the Paleolithic sites dated by our team, and red symbols indicate the research of other teams; MPT, the Mid-Pleistocene transition; GPTS, geomagnetic polarity time scale; 1. Longgangsi-3; 2. Yuelianghu; 3. Wutaicun; 4. Guanmenyan; 5. Shangbaichuan; 6. Wujiagou; 7. Liubeijiuchang; 8. Longgangsi-2; 9. Miaokou; 10. Qiaojiayao; 11. Yaochangwan; 12. Luojiacun; 13. Longgangsi-1; 14. Liuwan; 15. Jijiawan; 16. Ganyu; 17. Longyadong Cave; 18. Dali; 19. Changyang Cave; 20. Houfang; 21. Hejialiang; 22. Dishuiyan; 23. Liangshancun; 24. Lianfeng-1; 25. Lianfeng-2; 26. Yangjiaping; 27. Erligou; 28. Chihe-1; 29. Chihe-2; 30. Qujiahe; 31. Jinkuangcun-2; 32. Jinkuangcun-3; 33. Luohansi-2; 34. Jigongshan; 35. Jinshuihekou; 36. Yanling; 37. Shizilukou; 38. Huaishuping; 39. Huoshiwa; 40. Houshanpo; 41. Diaozhai; 42. Zhuangzixun; 43. Luohansi-1; 44. Xihoudu; 45. Shangchen; 46. Gongwangling; 47. Yunxian; 48. Meipu; 49. Shuigou-huixinggou; 50. Chenjiawo; 51. Bailongdong Cave; 52. Shuangshu; 53. Huanglong Cave; 54. Maling 2A; 55. Beiyao).

Before 1.2 Ma, only two hominin occupation records have been found in the northern QMR at present, and none of the records has been found in the South Luohe River valley in the eastern QMR and the Hanjiang River valley in the southern QMR.

During 1.2–0.7 Ma, the number of the hominin fossils and Paleolithic sites began to increase gradually, and the early hominins began to occupy the Hanjiang River valley and the South Luohe River valley. It is worth noting that hominin occupations were mainly concentrated in the eastern and southern QMR, while only few records (in the sites with independent ages) have been found in the Lantian Basin in the northern QMR. We speculate that this may be related to the MPT, and a detailed explanation can be found in Section 3.3.

From 0.7 Ma, the number of the sites continued to increase on a large scale. The records of the early hominin occupation increased in the northern QMR, and the scale of the hominin occupation in the eastern and southern QMR further expanded. This outcome may be related to the largest and longest-lasting warm and humid event of the S5 layer (MIS 13, 14, 15) since the Quaternary, which could have provided the optimal environment for hominin occupation and settlement. Hao et al. (2015) [115] suggest that the extra-long Northern Hemisphere interglacial climate during MISs 15−13 provided favorable conditions for the second major dispersal episode of African hominins into Eurasia. Around 0.2 Ma, climate change was mainly dominated by the warm and humid interglacial period, especially the Last Interglacial Period, S1 (MIS 5), at about 0.1 Ma [7,116,117], which was

suitable for hominin survival. On the other hand, the adaptability of the hominins to the environment may have also been gradually enhance. Therefore, hominin settlement and dispersal were sustained with greater intensity and a wider range after 0.7 Ma.

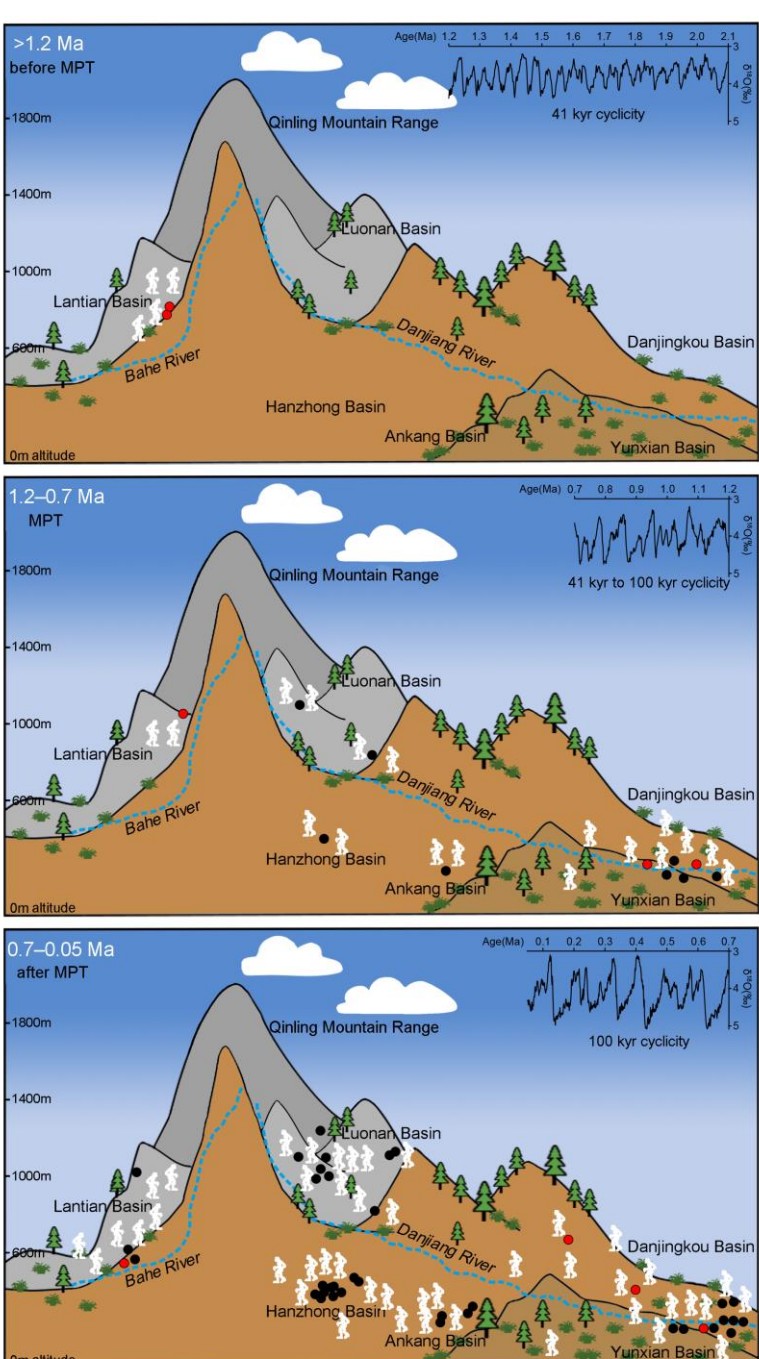

**Figure 4. Spatiotemporal distribution characteristics of hominin migration and dispersal in the QMR**. The marine oxygen isotopes indicate the changes in the dominant periodicity of the climatic cycles (cyclicity) [106]. The black dots represent the hominin fossil and Paleolithic sites dated by our team, and the red dots represent the research by other workers. MPT, the Mid-Pleistocene transition).

### 3.3. The Mid-Pleistocene Transition Drove Early Hominin across the QMR

The "Mid-Pleistocene transition (MPT)" was an important global climate event, which was mainly manifested in the change of the dominant periodicity of the glacial-interglacial cycles [112,114], the significant increase in global ice volume [118], and the significant increase in the global land aridification process [119–124]. Prior to the MPT, the climate was

characterized by relatively stable cycles (approximately 41 kyr) of warm and cold periods. By contrast, during the MPT, the dominant periodicity of the climatic cycles (cyclicity) changed from 41 kyr to 100 kyr [125] (Figure 4), resulting in more extreme and prolonged glacial periods, such as the two cold and dry glacial periods that occurred at 1.162–1.080 Ma and 0.952–0.865 Ma, respectively [126].

Scholars have proposed that the migration and dispersal of early hominins in China were related to the MPT [2,7,10,11,19,95]. Sun et al. (2018) proposed that, driven by the MPT, climate change in the glacial–interglacial period intensified, and the southern QMR became a "refuge" for the hominins [7]. Recent studies on leaf wax isotopes in the Eastern Qinling Mountains have further supported this viewpoint [127]. Yang et al. (2020, 2021) found that the geographical distribution of hominins changed across the MPT, showing a trend of southward movement [10,11].

How did the MPT affect the course of early hominin occupation and the development of hominin adaptations? The MPT drove early hominin migration and dispersal by affecting terrestrial ecosystems [2,7,9–13,19,95]. Studies show that the climate became dry and cold, with forest degradations and expansions of grasslands and deserts in northern China during the MPT [128,129]. After 1.2 Ma, the vegetation composition in northern China of alternating warm temperate forest and temperate forest was replaced by alternations of forest and grassland [128]. Changes in vegetation could then affect the species and distribution of Quaternary mammalian groups, and may have accelerated the rate of turnover of the large mammalian fauna [129–131]. The changes in terrestrial ecosystem caused by climate change would have affected the living environment of early hominins. Under the dominance of longer and colder glacial climates, the increased extreme living environmental pressure may have led to the southward diffusion of northern hominin populations.

As seen in Figures 3 and 4, hominin occupation records from before the MPT were discovered in the northern QMR, but not in the southern and eastern QMR. By contrast, the hominin occupation in the northern QMR was significantly less than that in the southern and eastern QMR during the MPT. In addition, the first appearance of lithic artifacts in the southern QMR is in the L15 layer (MIS 38) (ca. 1.22–1.19 Ma) at Longgangsi-3, which could represent the start of the MPT. As a result, we speculate that, under the influence of the MPT, the ecological structure and resources of the northern QMR deteriorated, with the climate becoming dry and cold, and thus, this was not a place for permanent year-round settlement for the hominins during the longer and more severe glacial periods. Therefore, early hominins made corresponding responses and adaptations and migrated to the south, where conditions were more suitable for survival. Our team proposes that, driven by the MPT, climate change in the glacial–interglacial period intensified, and the southern QMR became a "refuge" for the hominins [7].

Although the Qinling Mountains have the function of isolation and act as a barrier to the climate in the north and south of China, this was not an obstacle to hominin migration and diffusion between the north and the south [17]. The hominins could cross the QMR through a large number of river networks and exposed river beds. The range of the early human dispersal in the northern QMR reached about 800 m above sea level. They just needed to climb along the Bahe River valley to about 1400 m, that is, climb 600 m to cross the Qinling Mountains, reach the eastern QMR, and then reach the Danjiangkou Basin via the Danjiang River valley. Our team has proven that the Danjiang River Valley, as a link connecting the Bahe River, South Luohe River, and Hanjiang River Valleys, may have been one of the main corridors for hominins to cross the QMR, driven by the MPT [19]. It should be emphasized that it was not that there was no hominin occupation and settlement during the glacial periods in the northern QMR. The crossing and diffusion were only the traction of changes in the ecological environment and resources, which caused the diffusion and migration of some hominin groups.

## 4. Conclusions

During the Pleistocene, a large area of loess was deposited in the QMR, thus preserving a rich record of Paleolithic cultural remains, making it possible to reconstruct the early hominin occupation, migration, and dispersal and explore the role of climate therein [7,15,17]. Our team has conducted stratigraphic and dating studies on 44 hominin fossils and Paleolithic sites. Combined with published ages from other sites, we have tried to establish the chronological sequence of the Paleolithic sites and study the mechanism of climate impact on hominin occupation and dispersal in the QMR. According to the spatiotemporal distribution characteristics of the Paleolithic sites in the northern, eastern, and southern QMR, we have identified five stages of the hominin occupation. Before 1.2 Ma, early human occupation appeared sporadically, mainly in the northern QMR; from c 1.2–0.7 Ma, hominin settlement and dispersal began to occur widely, and appeared successively in the southern and eastern QMR; from c 0.7–0.3 Ma, the scale of hominin activity expanded, more particularly in the interglacial periods, MIS 9, 11, 13, 14, 15; during c 0.3–0.05 Ma, hominin occupation expanded yet further, and the highest number of sites was discovered, which were mainly distributed in the southern and eastern QMR, and hominin settlements and dispersals were most active in the glacial period, L2 (MIS 6), and interglacial period, S1 (MIS 5); from c 0.05–0.01 Ma, the records of hominin occupations decrease again. There may be a certain correlation between the MPT and early hominin occupation, migration, and dispersal in the QMR. We suggest that ecosystem changes caused by the MPT drove early hominins to migrate southwards across the QMR and that the southern QMR was a glacial refuge.

Although constrained by factors such as the number of Paleolithic sites, uncertainties in dating methods, etc., the chronological data presented here reflect the major spatiotemporal distribution characteristics of the hominin occupation and the driving mechanism of early hominin dispersal in the QMR. Ongoing research in the region will provide richer and more comprehensive data, which will facilitate the establishment of a more robust chronological sequence of hominin occupation, and deepen the study of the relationship between climate change and hominin migration and dispersal.

**Author Contributions:** Conceptualization, methodology, and validation, X.G. and X.S.; data sources, H.L., S.W. and C.L.; data collection and analysis, X.G.; writing original draft preparation, X.G.; writing review and editing, X.S., H.L. and S.W. All authors have read and agreed to the published version of the manuscript.

**Funding:** This research was funded by the National Natural Science Foundation of China, grant number 41972185 and the Major Program of National Social Science Foundation of China, grant number 19ZDA225.

**Data Availability Statement:** The data presented in this study are available upon request from the authors.

**Acknowledgments:** We thank Xu Xinghua, Lu Yiming and Lu Ying for their help in collecting and sorting Paleolithic site data. We are very grateful to Peng Zhang and Micheal Meadows for their modification suggestions. And we are sincerely grateful to the editors and referees for their valuable comments and suggestions.

**Conflicts of Interest:** The authors declare no conflict of interest.

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
