# Peer review of "Early Hominin Dispersal across the Qinling Mountains, China, during the Mid-Pleistocene Transition"

_land, doi:10.3390/land12101882_

Round 1

Reviewer 1 Report

This study presents a possible dispersion pattern of Hominin during the Paleolithic in the Qinling Mountain Range. The analysis was mainly based on the chronological order of the cultural sites in the Qiling Mountain Range during the Paleolithic. The paper also discussed the mechanism which shaped the pattern of the dispersion.

Generally, this paper is well organized and presented with many dating information. However, there are several uncertainties concerning the reliability of the study.

1. As the authors mentioned, there are several sites with different dating results. How do you choose the one that you use for this paper?

2. The chronological sequence of the sites is essential for this study and I suggest you present a table of the sites and the dating results, as well as the dating method. 

3. This study included several dating methods and I suggest you make a clear description of the dating method that you have conducted. For instance, for OSL dating, please present how and where or even in which layer you collected the samples, and how the experiment was conducted.    

At last, the discussion part three (4.3) is a bit weak. I suggest making a stronger statement for that section. 

I am not very qualified to judge the English fluency since i am not a native speaker. However, from my point of view, i believe, the English language could be improved with minor revisions. 

Author Response

1. Summary

Thank you very much for taking the time to review this manuscript. Please find the detailed responses below and the corresponding revisions in the re-submitted files.

2. Point-by-point response to Comments and Suggestions for Authors

Comments 1: As the authors mentioned, there are several sites with different dating results. How do you choose the one that you use for this paper?

Response 1: The chronology of the Paleolithic sites in the QMR summarized in this paper mainly serves to establish a chronological framework for hominin occupation and dispersal. We construct this framework on a longer time scale, such as >1.2 Ma, 1.2-0.7 Ma, 0.7-0.3 Ma, 0.3-0.05 Ma and 0.05–0.01 Ma. Although some Paleolithic sites have different chronological results, they are basically included in the time framework we have constructed, so there is no need to choose from them. For example, different scholars obtained six different age results at the Yunxian site, including 0.87-0.83 Ma, 0.936 Ma, 0.581 Ma, 0.8-0.785 Ma, 1.0-0.78 Ma, and 1.1 Ma, most of which belong to the stage of 1.2-0.7 Ma.

Comments 2: The chronological sequence of the sites is essential for this study and I suggest you present a table of the sites and the dating results, as well as the dating method.

Response 2: We fully agree with you. We have added a table of the chronological results of the 55 hominin fossils and Paleolithic sites in the “Chronological study” section on page 5.

Comments 3: This study included several dating methods and I suggest you make a clear description of the dating method that you have conducted. For instance, for OSL dating, please present how and where or even in which layer you collected the samples, and how the experiment was conducted.

Response 3: The ages of all the Paleolithic sites included in this manuscript are already published. We use those data to establish a chronological framework for the hominin occupation and dispersal. Therefore, we only provide a brief description of the age of each site and the dating methods used, without going into further details such as sampling, experiment, etc.

Comments 4: the discussion part three (4.3) is a bit weak. I suggest making a stronger statement for that section. 

Response 4: Thank you for pointing this out. We have added some related statements in the discussion section on page 13, as follows:

“Scholars have proposed that the migration and dispersal of early hominins in China are related to MPT [2, 7, 10-11, 19, 95]. Sun et al. (2018) proposed that, driven by the MPT, climate change in the glacial-interglacial period intensified and the southern QMR became a “refuge” for the hominins to live sustainably [7]. Recent studies on Leaf wax isotopes in the Eastern Qinling Mountains have further supported this viewpoint [127]. Yang et al. (2020, 2021) found that the geographical distribution of hominins changed across the MPT, showing a trend of southward movement [10, 11].”

3. Response to Comments on the Quality of English Language

Point: I am not very qualified to judge the English fluency since I am not a native speaker. However, from my point of view, I believe, the English language could be improved with minor revisions. 

Response: Thank you for pointing this out. We have carefully checked and corrected the issues regarding the use of the English language.

4. Additional clarifications

Thank you again for your comments and suggestions. We have made efforts to revise the parts of the manuscript that need improvement

Reviewer 2 Report

I have added the text with a few revisions/corrections which I think  would be important to make (in yellow). In my opinion a few more references should be added in a few key point A  revision of the English language by a native speaker would improve greatly the text. Also te summary should be slightly improved

It should be revised by a native speaking colleague. There are a few minor issues. However the English text is comprehensible, though not perfect. It would be important to improve it

Author Response

1. Summary

Thank you very much for taking the time to review this manuscript. Please find the detailed responses below and the corresponding revisions in the re-submitted files.

2. Point-by-point response to Comments and Suggestions for Authors

Comments: I have added the text with a few revisions/corrections which I think would be important to make (in yellow). In my opinion, a few more references should be added in a few key point. A revision of the English language by a native speaker would improve greatly the text. Also, the summary should be slightly improved

Response: We greatly appreciate your detailed modification suggestions. We have made corresponding modifications and added the modified paper in the attachment.

The parts of the manuscript that were not expressed clearly due to inaccurate wording have been revised, such as “abundant hominin fossils and Paleolithic cultural relics” in the abstract has been modified to “more than 500 hominin fossils and Paleolithic sites”, “rich cultural relics” in the abstract has been revised to “rich cultural remains”, “ages of 55 dated hominin fossils 18 and Paleolithic sites” in the abstract has been revised to “chronology of 55 dated hominin fossils and Paleolithic sites”, “Paleolithic cultural relics” in section 2.1 has been modified to “Paleolithic cultural remains”, “Sites occurring during the Early Pleistocene include Gongwangling, Shangchen and Xihoudu” in section 2.2.1 has been modified to “The sites belonging to the Early Pleistocene include the Gongwangling, Shangchen, and Xihoudu”, and so on. We have added some related references in a few key point. For example, the reference “Sun, X.J.; Wang, P.X., How old is the Asian monsoon system? —Palaeobotanical records from China. Palaeogeography, Palaeoclimatology, Palaeoecology 2005, 222, 181-222” was added in “The northern QMR is characterized by a temperate monsoon and temperate continental climate, while the southern and eastern QMR are characterized by a subtropical monsoon climate [55]” (in section 2.1). In addition, we have made modifications to the summary and also improved the English language.

3. Response to Comments on the Quality of English Language

Point: It should be revised by a native speaking colleague. There are a few minor issues. However, the English text is comprehensible, though not perfect. It would be important to improve it 

Response: Thank you for pointing this out. We have carefully checked and corrected the issues regarding the use of the English language.

4. Additional clarifications

Thank you again for your comments and suggestions. We have made efforts to revise the parts of the manuscript that need improvement
